Different intensities of basketball drills affect jump shot accuracy of expert and junior players

Marcolin Giuseppe 1
Camazzola Nicola 2
Panizzolo Fausto Antonio 3
Grigoletto Davide 1
Paoli Antonio antonio.paoli@unipd.it 1
1 Department of Biomedical Sciences, University of Padova , Padova , Italy
2 School of Human Movement Sciences, University of Padova , Padova , Italy
3 John A. Paulson School of Engineering and Applied Sciences, Harvard University , Cambridge , MA , United States of America
Keogh Justin
Electronic publication date: 2018 Feb 14
Publication date: 2018
Volume: 6
Electronic Location ID: e4250
Received 2017 Sep 13; Accepted 2017 Dec 19
Copyright: ©2018 Marcolin et al.
Copyright year: 2018
Copyright holder: Marcolin et al.
License: This is an open access article distributed under the terms of the Creative Commons Attribution License, which permits unrestricted use, distribution, reproduction and adaptation in any medium and for any purpose provided that it is properly attributed. For attribution, the original author(s), title, publication source (PeerJ) and either DOI or URL of the article must be cited.
License URL: https://creativecommons.org/licenses/by/4.0/

Keywords: Sport performance, Shot accuracy, Level of exertion, Technical drills

Funding: The authors received no funding for this work.

==============================
Background

In basketball a maximum accuracy at every game intensity is required while shooting. The aim of the present study was to investigate the acute effect of three different drill intensity simulation protocols on jump shot accuracy in expert and junior basketball players.

Materials & Methods

Eleven expert players (age 26 ± 6 yrs, weight 86 ± 11 kg, height 192 ± 8 cm) and ten junior players (age 18 ± 1 yrs, weight 75 ± 12 kg, height 184 ± 9 cm) completed three series of twenty jump shots at three different levels of exertion. Counter Movement Jump (CMJ) height was also measured after each series of jump shots. Exertion’s intensity was induced manipulating the basketball drills. Heart rate was measured for the whole duration of the tests while the rating of perceived exertion (RPE) was collected at the end of each series of shots.

Results

Heart rate and rating of perceived exertion (RPE) were statistically different in the three conditions for both expert and junior players. CMJ height remained almost unchanged in both groups. Jump shot accuracy decreased with increasing drills intensity both in experts and junior players. Expert players showed higher accuracy than junior players for all the three levels of exertion (83% vs 64%, p < 0.001; 75% vs 57%, p < 0.05; 76% vs 60%, p < 0.01). Moreover, for the most demanding level of exertion, experts showed a higher accuracy in the last ten shots compared to the first ten shots (82% vs 70%, p < 0.05).

Discussion

Experts coped better with the different exertion’s intensities, thus maintaining a higher level of performance. The introduction of technical short bouts of high-intensity sport-specific exercises into skill sessions should be proposed to improve jump shot accuracy during matches.

Introduction

Basketball is a common sport where conditioning and fatigue affect performance (Erculj & Supej, 2009). Very specific physiological requirements are associated with this sport, with mean heart rate during live time of 169 ± 9 beats per minute and mean blood lactate concentration of 6.8 ± 2.8 mmol L−1 (McInnes et al., 1995). Together with a high level of fitness, maximum accuracy is required while performing specific motor tasks such as shooting at a target (Erculj & Supej, 2009).

Due to these specific characteristics several studies analyzed the effect of physiological fatigue as well as physical exertions at different intensities on several aspects of basketball. Ahmed (2013) investigated the effect of upper extremity fatigue on grip strength and passing accuracy, showing a significant decrease of both these variables after the fatigue protocol administered. Similarly, it has been found that a detriment in the passing performance among novice and expert basketball players followed a high intensity total body fatigue protocol (Lyons, Al-Nakeeb & Nevill, 2006). The effect of fatigue on three-point shooting has been investigated from a biomechanical point of view analyzing the position of the release arm and shoulder girdle showing that all the measured angles decreased drastically as a consequence of the heavy level of fatigue (Erculj & Supej, 2009). Three point-shooting accuracy has also been investigated after two different resistance circuit training protocols showing a reduction in accuracy only after the most intensive protocol (Freitas et al., 2016). Conversely, free throw shooting analysis demonstrated that fatigue did not affect kinematics and that shooting technique was the same during successful and unsuccessful shots (Uygur et al., 2010). Similarly repeated sprint test performance indices remained unchanged after full time or even improved after half time in comparison with the same indices performed after the warm up (Meckel, Gottlieb & Eliakim, 2009).

In the evaluation of fatigue on sport skill performance, it has been shown that a key point is the employment of sport-specific training methods to induce fatigue (Davey, Thorpe & Williams, 2002; Davey, Thorpe & Williams, 2003; Vergauwen et al., 1998). To the best of our knowledge, while in water polo (Royal et al., 2006), tennis (Lyons et al., 2013) and soccer (Rampinini et al., 2008) accuracy was evaluated employing these sport-specific methods, while in basketball passing (Ahmed, 2013; Lyons, Al-Nakeeb & Nevill, 2006) and shooting (Freitas et al., 2016) accuracy was quantified after standard strength training exercises.

Therefore the purpose of the present study was to analyze the effect of three different exertion’s intensities induced with sport-specific drills on the jump shot accuracy and jump height of expert and junior basketball players. Our hypothesis was to find in both groups a decrease of jump shot accuracy and jump height as exertion increased. Moreover we expected expert players to better cope with the three exertion’s intensities compared to junior players because of their higher level of expertise.

Materials & Methods

Participants

Twenty-one basketball players divided in two groups volunteered in the study. The first group included 11 expert players (age 26 ± 6 yrs, weight 86 ± 11 kg, height 192 ± 8 cm) and the second 10 junior players (age 18 ± 1 yrs, weight 75 ± 12 kg, height 184 ± 9 cm). Experts trained four times per week while junior players three times. Inclusion criteria included at least 10 years of competitive basketball experience for expert players and five years for junior players, regular participation to official matches and lack of any muscle and tendon pathologies to upper and lower limbs at the time of the study. A detailed description of the experimental procedures was given to each participant which provided an informed consent prior to testing. The study was approved by the ethical committee of the Department of Biomedical Sciences, University of Padova (number HEC-DSB11/16).

Experimental protocol

Each participant was asked to perform three series of twenty jump shots at three different exertion’s intensities. A 15 min standardized warm up, consisting of 10 min of aerobic conditioning and 5 min of free throw shots, took place before the three series of jump shots. In the low intensity protocol (LIP) the participant stayed with feet on the free throw line, received the ball from a teammate positioned under the basket, and performed the jump shot. The teammate moved to regain the ball to perform the successive pass. This procedure was repeated until the twenty jump shots were completed. In the moderate intensity protocol (MIP) each participant started the series from the half-court line. Then he ran slowly to one of the two cones placed on the three-point line, by the side of the extension of the long sides of the 3 s area. After reaching the cone, he performed a side cutting maneuver towards the free throw line where he received the ball from a teammate and performed the jump shot. Subsequently the participant returned to the half-court line walking and started again. This course was repeated until the twenty jump shots were completed. Unlike the moderate intensity, in the high intensity protocol (HIP) participants sprinted towards the cone and, after the side cutting maneuver, towards the free throw line. The return to the half-court line was done half walking and half slowly running. In both moderate and high intensity protocols each participant had to choose one of the two cones and then perform the side cutting maneuver on the same cone for all the trials. The rest among the three series was set to 4 min. Before the beginning of the first series of jump shots and immediately after the moderate and high intensity series three counter movement jumps (CMJ) were collected for each athlete by means of a Bosco conductance mat (Globus Italia, Treviso, Italy) which estimates the height of the jump measuring the flight time. For the whole duration of the tests participants wore a chest band which recorded the heart rate at a sampling rate of 1 Hz (Garmin, Kansas City, KS, USA). In order to calculate Karvonen heart rate reserve (%HRR) (Karvonen, Kentala & Mustala, 1957) at each intensity of induced exertion, resting heart rate was assessed by each participant when they woke up in the morning for the two days prior to the tests. At the end of each set of shots the rating of perceived exertion (RPE) was recorded by means of a 6–20 Borg scale (Haile, Gallagher & Robertson, 2015). A schematic of the experimental setup is reported in Fig. 1.

Figure 1 Experimental protocol.

Schematic representation of the experimental protocol.

Statistical analysis

One way repeated measurements analysis of variance (ANOVA) was used to compare the three exertion’s intensities. Significant level was set at p < 0.05. If the statistical significance was reached, Tukey’s multiple comparison test was employed. Unpaired t-test was used to compare expert and junior players at every level of induced exertion. Data analysis was performed with the software package GraphPad Prism version 4.00 for Windows (GraphPad Software; San Diego California, CA, USA). Statistical effect size was calculated with the G*Power 3.1.5 software (Faul et al., 2007).

Results

Significant differences among the exertion’s intensities reached during the three sets of jump shots for the two groups of athletes were reported by the statistical analysis (Table 1). Among experts, heart rate values were lower in LIP with respect to MIP (p < 0.01, ES: 1.00) and HIP (p < 0.01, ES: 3.06) and the rate of perceived exertion was lower in LIP with respect to MIP (p < 0.01, ES: 1.50) and HIP (p < 0.01, ES: 4.00). Among junior players, heart rate values were lower in LIP with respect to MIP (p < 0.01, ES: 0.75) and HIP (p < 0.01, ES: 2.19) and the rate of perceived exertion was lower in LIP with respect to MIP (p < 0.05, ES: 1.09) and HIP (p < 0.01, ES: 3.56). In the LIP condition, RPE was higher for juniors players in comparison with experts players (p < 0.05, ES: 1.25). MIP and HIP conditions were perceived equally demanding by experts and junior players with no statistically significant differences among RPE values. Shot accuracy of expert players was 83% (LIP), 75% (MIP) and 76% (HIP), while for the group of junior players it was 64% (LIP), 57% (MIP), 60% (HIP). Conversely, jump height was very similar for junior players and expert players (Table 1). Moreover, a statistically significant increase of expert players’ jump height after HIP with respect to warm up was found (p < 0.01; ES = 0.58). The analysis of jump shot accuracy between expert and junior players revealed a better proficiency among expert players at every intensity of induced exertion (LIP: p < 0.001, ES: 1.74; MIP: p < 0.05, ES: 1.03; HIP: p < 0.01, ES: 1.37), as reported in Fig. 2. A further analysis compared the effect of the three protocols on jump shot accuracy of the first ten shots with the last ten shots. Expert players showed a decrease of accuracy only between LIP and HIP (p < 0.05, ES: 1.03) while junior players showed no statistically significant differences in the first ten shots. On the other hand, variation of the accuracy of the last ten shots was not statistically significant neither in expert nor in junior players. Moreover, expert players showed a higher accuracy comparing the second ten shots with the first ten shots only in the HIP condition (p < 0.05, ES: 0.73). No statistically significant differences were detected among junior players. All the results relative to the first and to the last ten shots for both groups are reported in Table 2.

Table 1 Different intensities of basketball drills affect jump shot accuracy of expert and junior players.

Heart rate, rate of perceived exertion (RPE), jump shot accuracy and jump height data. Heart rate, rate of perceived exertion (RPE), jump shot accuracy and jump height recorded immediately after each fatigue protocol. Data are presented as mean ± s.

	Fatigue protocol	
	LIP	MIP	HIP	
Heart Rate (beats min−1)				
Expert players	116 ± 14	129 ± 11a	154 ± 9a,b	
Junior players	134 ± 16	145 ± 11a	165 ± 9a,b	
Karvonen Heart Rate reserve (%)				
Expert players	47 ± 9	57 ± 7a	75 ± 5a,b	
Junior players	57 ± 12	65 ± 9a	79 ± 7	
Rating of Perceived Exertion (6–20)				
Expert players	8 ± 1.5	10.2 ± 1.3a	13.5 ± 1.1a,b	
Junior players	9.8 ± 1.3	11.2 ± 1.2a	14.4 ± 1.3a,b	
Jump shot accuracy (baskets scored)				
Expert players	16.6 ± 1.6	15.1 ± 2.8	15.2 ± 2.2	
Junior players	12.8 ± 2.7	11.4 ± 4.2	12 ± 2.4	
Jump height (cm)				
Expert players	49.1 ± 3.4	49.8 ± 3.5	50.9 ± 2.8a	
Junior players	46.1 ± 2.1	47.7 ± 3.8	46.9 ± 3.4	
Notes.

a Different from low intensity fatigue protocol (LIP).

b Different from moderate intensity fatigue protocol (MIP).

Figure 2 Jump shot accuracy.

Baskets scored in the LIP (A), MIP (B) and HIP (C) conditions: comparison between experts and junior players (* p < 0.05; ** p < 0.01; *** p < 0.001).

Table 2 Different intensities of basketball drills affect jump shot accuracy of expert and junior players.

Expert and junior shoot performance. Baskets scored for each level of induced fatigue.

	LIP	MIP	HIP	
Series of shot	1–10	11–20	1–10	11–20	1–10	11–20	
Groups							
Expert players	8.5 ± 1	8.1 ± 1.1	7.6 ± 1.7	7.5 ± 1.7	7 ± 1.8	8.2 ± 0.6a	
Junior players	6.3 ± 2	6.5 ± 1.6	5.5 ± 2.4	5.9 ± 2.2	5.9 ± 1.3	6.1 ± 1.5	
	
Notes.

a Experts: Different from 1–10 series.

Discussion

To assess the effect of different exertion’s intensities on sport-specific skills it is essential to employ sport-specific protocols for an ecological validity of the experimental results (Lyons, Al-Nakeeb & Nevill, 2006; Royal et al., 2006). In the present study, three different exertion’s intensities (LIP, MIP and HIP) were induced by means of basketball drills such as sprinting, cutting maneuvers and passes allowing to investigate jump shot accuracy in expert and junior players. A decrease in shot accuracy was detected for both groups similarly to previous investigations where technical skills deteriorated as a consequence of fatigue (Freitas et al., 2016; Lyons, Al-Nakeeb & Nevill, 2006; Rampinini et al., 2008). Conversely, the results of the present study showed no differences in jump height after the three protocols. Similar results were obtained investigating repeated sprint tests at different game stages (Meckel, Gottlieb & Eliakim, 2009) underlying the importance of an intense warm up to improve sprint performance in the initial phases of a basketball match. Despite the RPE and HR results, in the present work the three different exertion’s intensities proposed cannot be considered as fatigue promoters since a decrease of the jump performance was not detected after MIP and above all HIP. However, the intensities proposed with regards to HIP allowed to investigate how the jump shot accuracy varied in a contest similar to a real match play.

Therefore our results supported the suggestion of training at level of exertion similar to those recorded during competitive games (Erculj & Supej, 2009; Freitas et al., 2016; Lyons et al., 2013; Lyons, Al-Nakeeb & Nevill, 2006) together with the recommendation to employ sport-specific drills to increase the exertion’s intensity (Lyons, Al-Nakeeb & Nevill, 2006; Royal et al., 2006). The decrease in jump shot accuracy was indeed more marked for both groups comparing LIP with HIP rather than MIP compared to HIP. Thus, an increased number of high intensity skill sessions should aim to improve the shot accuracy at those exertion’s intensities representative of a match. However, during the introduction of these technical short bouts of high-intensity exercise into skill sessions the technique should be monitored to assure a correct execution (Lyons et al., 2013).

The comparison between expert and junior players showed a higher accuracy in the expert group at each exertion’s intensity. Our results are comparable with previous investigations on basketball (Lyons, Al-Nakeeb & Nevill, 2006) and tennis (Lyons et al., 2013), showing how expert players can cope better with higher exertion’s intensities, thus maintaining a higher level of performance. This could be due to the fact that technical and motor patterns are strongly formed in experts and that they can adjust motor coordination strategies as a reaction to induced exertion better than junior players (Aune, Ingvaldsen & Ettema, 2008). Therefore it is clear again how with junior players the introduction of technical short bouts of high-intensity exercise into skill sessions should be carefully implemented to avoid technique alterations as much as possible (Lyons et al., 2013).

An additional interesting point relative to HIP is the higher shot accuracy recorded by expert players in the last ten shots with respect to that recorded in the first ten shots. Since athletes begin the HIP bout after a 4-minute recovery (see Fig. 1), only in the second part of the bout heart rate values were similar to those recorded in a real match (Erculj & Supej, 2009). In fact expert player mean heart rate was 143 ± 9 beats min−1 during the first ten shots and 165 ± 9 beats min−1 in the last ten shots. This condition, together with the employment of basketball-specific tasks, could have created sensory states similar to those experienced in contest inducing participants to use the same processes responsible for their expertise in match-play (Royal et al., 2006). For the same reason, junior players could have maintained their shot accuracy in the last ten shots in comparison with that recorded in the first ten shots. In fact their mean heart rate was 153 ± 10 beats min−1 and 176 ± 8 beats min−1 during the first and the last ten shots respectively. To this extent, we can hypothesize that in the last ten shots of HIP the decrease of shot proficiency due to their lower ability in coping with high exertion’s intensity was counterbalanced by the sensory states typical of a match-play they experienced.

Jump shot accuracy results referred to the highest exertion’s intensity excluded a possible relationship between shot proficiency and mental fatigue (Boksem, Meijman & Lorist, 2005; Faber, Maurits & Lorist, 2012). In fact, if mental fatigue would have been induced, the shot accuracy of the last ten shots in the HIP should have been the worst. Moreover it has been shown how mental fatigue affected marksmanship judgment in soldiers but not their shot accuracy (Head et al., 2017).

On the importance of the exertion’s intensity on performance an additional aspect deserves consideration. Previous literature (Meckel, Gottlieb & Eliakim, 2009) reported that an intensive warm up is needed to increase repeated sprint performance at the initial phases of the match. Our findings on the accuracy during HIP, comparing the first ten shots with the second ten shots, are in agreement with this theory. When heart rate was stabilized at a high value, accuracy increased in experts and stayed constant in junior players. Therefore, an intensive warm-up could be useful to activate a game-specific arousal since the initial phases of the match. Moreover, short intensive exercises could be proposed for the cases in which a player comes off the bench to reactivate this game-specific arousal entering in the court.

Conclusions

Our study showed how expert basketball players coped better with different exertion’s intensities with respect to junior players, thus maintaining a higher level of jump shot proficiency. Moreover, expert players showed at HIP the best shot accuracy when heart rate was high in the last ten shots. This high exertion’s intensity together with the employment of sport specific drills could have induced participants to activate during training the same processes responsible for their expertise in real match-play. Therefore our findings could be of practical interest for coaches to improve the efficacy of technical skill sessions during training and warm up before matches. However, the introduction of these high-intensity technical exercises into skill sessions should be carefully monitored to avoid technique alterations. Further studies are required to investigate how the exertion’s intensity of different sport-specific protocols could be used to activate game-specific arousal on athletes with distinct levels of expertise without altering the technical execution of the jump shot.

Supplemental Information

Data S1 Raw data

Heart rate, rate of perceived exertion (RPE), jump shot accuracy and jump height raw data.

Click here for additional data file.

Additional Information and Declarations

Competing Interests

Author Contributions

Human Ethics

Data Availability

The authors declare there are no competing interests.

Giuseppe Marcolin conceived and designed the experiments, performed the experiments, analyzed the data, wrote the paper, prepared figures and/or tables, reviewed drafts of the paper.

Nicola Camazzola performed the experiments, analyzed the data.

Fausto Antonio Panizzolo wrote the paper, reviewed drafts of the paper.

Davide Grigoletto performed the experiments, contributed reagents/materials/analysis tools, reviewed drafts of the paper.

Antonio Paoli conceived and designed the experiments, analyzed the data, wrote the paper, reviewed drafts of the paper.

The following information was supplied relating to ethical approvals (i.e., approving body and any reference numbers):

The study was approved by the ethical committee of the Department of Biomedical Sciences, University of Padova (number HEC-DSB11/16).

The following information was supplied regarding data availability:

The raw data is provided as a Supplemental File.

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
