# Peer review of "Different intensities of basketball drills affect jump shot accuracy of expert and junior players"

_PeerJ, doi:10.7717/peerj.4250_

## Round 0.1 · original submission · Major Revisions

The authors have presented some interesting results in this study, but both reviewer one and myself have some major concerns with some aspects of study design and the interpretation around the concept of fatigue. I therefore suggest you look over the reviewers comments, particularly reviewer one who made some excellent points regarding central verse peripheral fatigue; how some of the outcome measures such as the vertical jump scores don't necessarily match the differing levels of fatigue you have stated; as well as some of the other issues relating to the fatigue task that you provided the athletes. You will need to take on board these comments in your revisions if you wish to be published in this journal

Reviewer 1 ·

Basic reporting

No comment

Experimental design

The authors have not defined ‘fatigue’ as either peripheral or central, so accordingly, cannot postulate that fatigue has actually been induced. Yes, the player’s heart rate and perceived exertion may have increased, but this is not because of heightened levels of fatigue – reinforced by the fact that the expert players jump height actually increased after the high fatigue protocol! Given this, I am almost certain neuromuscular fatigue was not induced, making the entirety of their results and discussion misguided.

The protocols used to ‘fatigue’ the players appeared nonsensical, and lacked construct validity. Why did the authors not use the protocols described by Scanlan et al. (2012 – The development of the basketball exercise simulation test)? This test was comprehensively developed, using local positioning system data extracted from game-play to inform its construct. Had the authors used this, it is likely they would have incurred ‘fatigue’.

Validity of the findings

The authors note a decline in performance of the junior group following the high fatigue protocol, citing the kinematics of the upper body being the confounding factor given the maintenance of jump height (again, showing the lack of fatigue actually induced). However, given no kinematic analysis was done, how can this be the author’s conclusion? More directly, I feel that the decline in shooting performance was more of a reflection of the attention shift participants are likely to have encountered given the perception of being ‘fatigued’ (see Maarten et al., 2005 – Effects of mental fatigue on attention: An ERP study).

Surely the expert players possessed a superior level of physiological fitness relative to the junior players? So, why did the authors not extract this data at the start and then modify the protocols to the relative physiological fitness of player groups? More directly, the ‘fatiguing’ protocol for the expert group could have simply been a warm up given the likely superior fitness they possess relative to the junior players, who are likely to have struggled more so with this protocol. Thus, are you not comparing apples with oranges?

Additional comments

The concept of fatigue and its impact upon jump shot accuracy in expert and junior basketball players is interesting. However, given the above points, I feel the study has just missed the mark here.

In addition to the more major commentary listed above, there are multiple minor points of concern throughout that the authors need to consider prior to re-submission.

Abstract
• The first sentence is nonsensical – what is conditioning and related fatigue?
Introduction
• The physiological parameters presented in Lines 39 and 40 hardly warrant the use of “Very severe”…. This is where I would be more inclined to focus on the movement demands of the players, as these are what would have impacted neuromuscular fatigue – making more sense for your experimental protocols.
• No use of paragraphs throughout?

Results
• Lines 122 – 123 – again, how can the authors state that fatigue has been induced? Jump height has maintained and actually increased slightly for the expert group! Further, the expert / junior differences are merely a reflection of training age, no?

General
• Reference inconsistencies / Figure and Table captions on different pages.

·

Basic reporting

There are a number of grammatical errors throughout the paper. I propose that the following changes be made for this paper to be accepted.

Abstract, Line 17- include a comma after basketball. Change to "In basketball, conditioning and related fatigue affect performance."

Abstract Line 17- include comma after moreover and remove "a". Change to "Moreover, maximum accuracy is required while shooting."

Abstract, Line 18- include "The" and change second "study" to investigate.
Change to "The aim of the study was to investigate the effect of three different levels of fatigue on jump shot...".

Introduction, Line 38- I would suggest to change severe to specific.

Introduction, Line 41- remove "a" before maximum. Change to "Together with a high level of fitness, maximum accuracy is required..."

Introduction, Line 46- change "Similarly, it has been showed a detriment...' to "similarly, it has been found that a detriment...".

Introduction, Line 48- change "3 points-shooting" to "3 point-shooting". Keep consistent throughout the rest of the article.

Introduction, Line 52- change worsening to "reduction in accuracy".

Introduction, Line 55- change "considering" to "during".

Introduction, Line 56&57- change "indices" to "scores".

Introduction, Line 56- you have misinterpreted the results of the repeated-sprint study. There was no difference reported between total sprint time, after the warm up and after the game. Only difference reported was sprint at half-time.

Introduction, Line 59- change "showed" to "shown"

Introduction, Line 59- include training in between sport-specific and methods. Change to "...sport-specific training methods..." . Keep consistent throughout the rest of the article.

Introduction, Line 68&69- Include the jump shot in this sentence. "Our hypothesis was that fatigue could induce in both groups a decrease of the accuracy and jump height as exertion is increased during the jump shot."

Introduction, Line 70- change "with respect to" to "compared with"

Experimental Procedure, Line 86- change "minutes" to "minute" and "consisting in" to "consisting of".

Experimental Procedure, Line 87- change "free shots" to "free throw shots". Keep consistent throughout the rest of the article.

Experimental Procedure, Line 90- change "proved" to "moved".

Experimental Procedure, Line 92- change "Then he had to run..." to "The participant ran..."

Experimental Procedure, Line 94- change "he" to "they".

Experimental Procedure, Line 94- include a comma, change to "After reaching the cone, they performed a cutting manoeuver..."

Experimental Procedure, Line 102- change "always on that for all the trials" to "the same for every trial."

Experimental Procedure, Line 106- change "fly time" to "flight time".

Experimental Procedure, Line 110- change "prior the tests" to Prior to the tests."

Experimental Procedure, Line 115- include significance instead of statistically or write both e.g. "If the statistical significance..."/ "If the significance...".

Results, Line 122- change “set’ to “sets”.

Discussion, Line 155- change “contribute” to contributor”.

Discussion, Line 177- I would suggest changing “proposed” to “implemented”.

Discussion, Line 203- change “experts” to “expert”.

Experimental design

No comment. procedure is satisfactory.

Validity of the findings

No comment. Validity of findings seemed satisfactory.

Additional comments

I think the procedure used was scientifically sound. It was a good approach to investigating the effect of fatigue on shooting accuracy and a well thought out study design.

I agree with the implications for skill training in basketball provided in the comments.

---

## Round 0.2 · accepted · Accept

The authors are to be congratulated on taking on board the comments of the two reviewers. As such, I am happy to let you know that your paper has been accepted for publication in PeerJ.

Reviewer 1 ·

Basic reporting

No comment

Experimental design

No comment

Validity of the findings

No comment

Additional comments

The authors should be congratulated for addressing both reviewer comments with respect. Their rebuttal was sound and where necessary, they addressed the reviewer commentary in a sound way - well done.

I have no further commentary and wish the authors luck for their future research ambitions.